# Lipopolysaccharide Stimulates A549 Cell Migration through p-Tyr 42 RhoA and Phospholipase D1 Activity

**DOI:** 10.3390/biom14010006

**Published:** 2023-12-20

**Authors:** Shohel Mahmud, Amir Hamza, Yoon-Beom Lee, Jung-Ki Min, Rokibul Islam, Oyungerel Dogsom, Jae-Bong Park

**Affiliations:** 1Department of Biochemistry, College of Medicine, Hallym University, Hallymdaehag-Gil 1, Chuncheon 24252, Kangwon-do, Republic of Korea; slm.btge@hallym.ac.kr (S.M.); amirhamza9@hallym.ac.kr (A.H.); barca9118@hallym.ac.kr (Y.-B.L.); jkmin0306@hallym.ac.kr (J.-K.M.); mrislam@btge.iu.ac.bd (R.I.); oyungerel.d@mnums.edu.mn (O.D.); 2National Institute of Biotechnology, Ganakbari, Ashulia, Savar, Dhaka 1349, Bangladesh; 3Department of Biotechnology and Genetic Engineering, Faculty of Biological Science, Islamic University, Kushtia 7003, Bangladesh; 4Department of Biology, School of Bio-Medicine, Mongolian National University of Medical Sciences, Ulaanbaatar 14210, Mongolia; 5Institute of Cell Differentiation and Aging, College of Medicine, Hallym University, Chuncheon 24252, Kangwon-do, Republic of Korea

**Keywords:** p-Tyr42 RhoA, PLD1, ZEB1, MYH9, cell migration

## Abstract

Cell migration is a crucial contributor to metastasis, a critical process associated with the mortality of cancer patients. The initiation of metastasis is triggered by epithelial–mesenchymal transition (EMT), along with the changes in the expression of EMT marker proteins. Inflammation plays a significant role in carcinogenesis and metastasis. Lipopolysaccharide (LPS), a typical inflammatory agent, promoted the generation of superoxide through the activation of p-Tyr42 RhoA, Rho-dependent kinase 2 (ROCK2), and the phosphorylation of p47phox. In addition, p-Tyr42 RhoA activated phospholipase D1 (PLD1), with PLD1 and phosphatidic acid (PA) being involved in superoxide production. PA also regulated the expression of EMT proteins. Consequently, we have identified MHY9 (Myosin IIA, NMIIA) as a PA-binding protein in response to LPS. MYH9 also contributed to cell migration and the alteration in the expression of EMT marker proteins. Co-immunoprecipitation revealed the formation of a complex involving p-Tyr42 RhoA, PLD1, and MYH9. These proteins were found to be distributed in both the cytosol and nucleus. In addition, we have found that p-Tyr42 RhoA PLD1 and MYH9 associate with the ZEB1 promoter. The suppression of ZEB1 mRNA levels was achieved through the knockdown of RhoA, PLD1, and MYH9 using si-RNAs. Taken together, we propose that p-Tyr42 RhoA and PLD1, responsible for producing PA, and PA-bound MYH9 are involved in the regulation of ZEB1 expression, thereby promoting cell migration.

## 1. Introduction

The migration of cells is vital in various biological phenomena, including embryogenesis, immune response, inflammation, morphogenesis, and wound healing. In particular, cancer cell migration is a critical step in metastasis, which is the main cause of cancer-related deaths. Epithelial–mesenchymal transition (EMT) has been related to the metastasis of cancer cells by promoting motility and invasion [1]. During the EMT process, the expression of proteins for adhering adjacent cells, including E-cadherin, Occludin, and Claudin, decreased, while N-cadherin, Vimentin, and EMT transcription factors such as Zeb1, Snail 1/2, and Twist are upregulated [1]. Here, our focus was on exploring the potential role of inflammation in EMT process within cancer cells. Accordingly, a typical inflammatory agent, LPS, was utilized for inducting inflammation in cancer cells. 

Lipopolysaccharide (LPS), a distinctive constituent found in the cell wall of Gram-negative bacteria, serves as a well-known activator of the inflammasome. Within intestinal epithelial cells, the activation of the inflammasome is triggered by LPS through the initiation of Toll-like receptor 4 (TLR4)-mediated signal transduction pathways [2]. Numerous reports found that LPS stimulates inflammatory responses, causing epithelial cells to change into cancerous forms. For example, it has been shown that LPS promotes the invasion and migration of prostate cancer (PCa) cells [3] and promotes epithelial–mesenchymal transition (EMT) through TLR4 stimulated by LPS [4]. 

LPS has been documented to trigger the activation of NADPH oxidase [5]. NADPH oxidase is comprised of membrane components, such as gp91phox and p22phox, as well as cytosolic components, including p40phox, p47phox, and p67phox. Upon cellular stimulation, cytosolic components, especially p47phox, undergo phosphorylation by several kinases. Subsequently, phosphorylated p47phox at the Ser345 residue is recruited to the membrane components, initiating the activation of NADPH oxidase. Simultaneously, Rac1/2-GTP, a member of the Rho GTPase subfamily, forms a complex with NADPH oxidase, significantly enhancing its activity [6]. 

Typically, Rho GTPase is involved in regulating a range of cellular processes, including the modulation of cytoskeletal proteins, cellular morphology, migration, and cell proliferation. The activities of Rho GTPases are regulated by specific factors such as guanine nucleotide exchange factors (GEFs), GTPase-activating proteins (GAPs), and the guanine nucleotide dissociation factor (GDI) [7]. The cell migration system encompasses extensively researched members of the Rho-GTPase family, including RhoA, Rac1, and Cdc42, which each play significant roles in regulating cell motility. RhoA specifically contributes to focal adhesion formation, actomyosin contraction, stress fiber formation, and retraction of the cell’s tail, thereby facilitating cell migration [8]. Studies have indicated that LPS can trigger the disruption of tight junctions in brain microvascular endothelial cells (BMECs) through the activation of the RhoA/ROCK signaling pathway [9].

In mammals, the intracellular enzyme phospholipase D (PLD) responds to external signals by hydrolyzing phosphatidylcholine (PC), the most abundant phospholipid in cell membranes, to generate the lipid second-messenger phosphatidic acid (PA). The resulting PA, produced by PLD, serves as a regulator for a wide range of cellular processes including vesicular trafficking, membrane fission and fusion, as well as cell growth, proliferation, and migration [10,11]. Two mammalian PLD, PLD1 and PLD2, have been identified, both of which include splice variants [12]. In both in vitro and in vivo contexts, the activity of PLD1 is significantly boosted by protein kinase C (PKC) as well as small GTPases such as the ADP-ribosylation factor (Arf) and Rho families [13]. In particular, the C-terminal domain of PLD1 interacts with the dominant active mutant RhoA G14V [14], and it was recently reported that the crystal structure of PLD1 provided an activation mechanism by RhoA and PI(4,5)P_2_ [15]. Notably, G protein-coupled receptors (GPCRs) activated nuclear PLD1 in vascular smooth muscle cells in response to lysophosphatidic acid (LPA) [16]. It is also reported that PLD1, but not PLD2, is upregulated in non-small cell lung cancer tissue [17]. Our previous study revealed that p-Tyr42 RhoA contributes to amplifying superoxide production and induces the cell migration of A549 lung cancer cell line in response to PMA [18]. Based on these reports, we investigated the function of p-Tyr42 RhoA in PLD1 activation, and its downstream signaling pathway. 

In this study, we observed that LPS promoted the p-Tyr42 RhoA levels, which activated PLD1 through the interaction between p-Tyr42 RhoA and PLD1 in A549 cells. LPS stimulated cell migration through the expression of EMT marker proteins, and particularly the promoter of which ZEB1 were associated with p-Tyr42 RhoA and PLD1 in the nucleus in response to LPS. Moreover, we discovered that MYH9, also known as Myosin IIA and NMIIA, was a PA-binding protein. 

## 2. Materials and Methods

### 2.1. Materials

The Tat-C3 recombinant fusion protein, composed of Tat-peptide and C3 toxin, was purified from *E. coli* using a His-Bind resin column (EMD Milliporre, #70239-3, Darmstadt, Germany). Y27632 (SCM075) was obtained from Millipore Sigma (Burlington, Burlington, NJ, USA). LPS (*E. coli* 055: B5), N-acetyl-L-cysteine (NAC, A7250), apocynin (A10809), Nonidet P-40 (NP-40), bovine serum albumin (BSA), and isopropyl β-D-thiogalactoside (IPTG) were procured from Sigma-Aldrich (St. Louis, MO, USA). Fetal bovine serum (FBS, 12484010), Dulbecco’s modified Eagle’s medium (DMEM, 11965092), and penicillin–streptomycin antibiotics (15140122) were obtained from GibcoBRL (New York, NY, USA). Protease inhibitor cocktail was purchased from ApexBio (Boston, MA, USA). Fetal bovine serum (FBS), penicillin, and streptomycin were purchased from Cambrex (Verviers, Belgium). LB Broth High Salt (MB-L4488) and skim milk powder (MB-S1667) were sourced from MBcell (SeoCho-Gu, Seoul, Republic of Korea). 4′6-diamidino-2-phenylindole (DAPI) was purchased from Invitrogen (Carlsbad, CA, USA). Alexa fluor-568 and Alexa fluor-594 reagents were obtained from Molecular Probes (Eugene, OR, USA). Protein A/G-agarose beads were purchased from Amersham Biosciences (Piscataway, NJ, USA). The JetPRIME DNA/si-RNA transfection reagent was obtained from Polyplus-transfection (Seoul, Korea). si-PLD1 (sc-44000), si-RhoA (sc-29471), and control si-RNA (sc-37007) were procured from Santa Cruz Biotechnology. si-MYH9 (Bioneer, #4627-1) was obtained from Bioneer (Daejeon, Republic of Korea). 

The antibodies utilized in this study were obtained from various providers. An antibody against phosphorylated Tyr42 Rho was generated through immunization with a peptide corresponding to p-Tyr42 Rho (epitope peptide T37VFEN (phospho-) Y42VADIE47). This peptide was synthesized from the phospho-Tyr42 precursor obtained from Young-In Frontier, Seoul, Korea. Normal IgG (sc:2025), actin (sc:58673), PLD1 (sc-28314), Lamin B (sc:965962), Tubulin (sc:32293), ROCK I (sc-17794), ROCK II, (sc:398519), RhoA (sc-418), N-cadherin (sc:393933), and Snail (sc:271977) were purchased from Santa Cruz Biotechnology (Dallas, TX, USA). MYH9 (14844-1-AP) was purchased from Proteintech (Rosemont, IL, USA). HA Tag (3724), p-Lamin A/C (2026) were purchased from Cell Signaling Technology. E-cadherin (ABP0083, Abbkine) and ZEB1 (ABP60963) were purchased from Abbkine (Wuhan, China). p-p47phox (#PA5-37806) was procured from Invitrogen (Waltham, MA, USA). Goat anti-rabbit and goat anti-mouse IgG conjugated with HRP were sourced from Enzo Life Sciences (Farmingdale, NY, USA).

### 2.2. Cell Culture

A549 cells, derived from human lung adenocarcinoma, were obtained from the Korean Cell Line Bank (Seoul, Republic of Korea). Subsequently, these cells were cultured in Dulbecco’s modified Eagle’s medium (DMEM) supplemented with 5% heat-inactivated fetal bovine serum (FBS) and antibiotics (100 U/mL of penicillin and 100 μg/mL of streptomycin) in a humidified environment with 5% CO_2_ at 37 °C. 

### 2.3. Superoxide Measurement

The levels of superoxide were assessed utilizing the fluorescent probe 2′,7′-dichlorofluorescin diacetate (DCFDA) cellular ROS assay kit (ab113851). A549 cells (2 × 10^5^) were stimulated with the specific treatment in serum-free media, followed by fixation using 4% formaldehyde for 15 min at room temperature. To induce fluorescence, the cells were treated with a solution of 5 μM DCFDA in 1 × PBS (500 μL) for 30 min at 37 °C in the dark. Fluorescence images were recorded using a fluorescence microscope (Axiovert 200, Zeiss; Göttingen, Germany) with filters configured for an excitation wavelength of 485 nm and an emission wavelength of 530 nm.

### 2.4. Transfection of Plasmid DNA and si-RNA

The A549 cells were transfected with plasmid DNA and small interfering RNA (si-RNA) using the JetPRIME reagent (polyplus transfection, Illkirch, France). Plasmid constructs HA-RhoA WT, HA-RhoA Y42E, HA-RhoA-Y42F and si-RNAs targeting RhoA, PLD1, MYH9, and control si-RNA were utilized in this study. In all si-RNA transfection experiments, A549 cells were initially seeded to attain 30–40% confluence in six-well culture dishes and subsequently transfected with si-RNAs (50–100 nM) using the JetPRIME transfection reagent by following the manufacturer’s instructions. When conducting both DNA and si-RNA transfection in the same culture dishes, si-RNA was transfected initially, followed by the transfection of DNA (2 µg plasmid DNA) after 24 h using fresh media containing 5% serum. After 48 h for DNA transfection and 72 h for si-RNA transfection, the cells were harvested, and protein expression levels were evaluated through Western blot analysis. 

### 2.5. Western Blotting

Western blot analysis was conducted on A549 cells. The cells were washed with 1 × PBS and lysed in RIPA buffer containing 20 mM Tris-HCl pH 7.4, 1 mM MgCl_2_, 1% (*v/v*) Nonidet-P40, and 125 mM NaCl. The lysates were supplemented with a phosphatase/protease inhibitor cocktail. After centrifugation for 15 min at 13,000× *g* at 4 °C, the protein concentration in the supernatants was determined using a BCA protein quantification assay kit (Abbkine, GA, USA). Subsequently, the cell lysates were separated using 10–14% SDS-PAGE and transferred to a PVDF membrane. Non-specific binding was mitigated by incubating the membrane with non-fat dried milk and 0.05% Tween-20 in a Tris-buffered saline solution. Specific antibodies were employed to probe the target proteins, followed by washing and incubation with peroxidase-conjugated secondary antibodies. Visualization of the bound antibodies was accomplished using an ECL system.

### 2.6. Immunoprecipitation (IP)

Immunoprecipitation was carried out in A549 cells. The cells were rinsed with 1 × PBS and lysed in a buffer containing 20 mM Tris pH 7.4, 120 mM NaCl, 1 mM MgCl_2_, and 1% Nonidet P-40. A protease/phosphatase inhibitor mixture was added to the lysates. The lysates were clarified through centrifugation and subsequently subjected to a preliminary clearance step using protein A/G-agarose beads for 1 h. The cleared supernatant was incubated with a specific antibody (1:1000 dilutions) overnight at 4 °C. Protein A/G-agarose beads (30 μL) were added to the lysate and the mixture was incubated with shaking. After the beads settled, mixtures were washed three times with a lysis buffer. The proteins bound to the beads were released and analyzed using immunoblotting with specific primary antibodies.

### 2.7. Cell Fractionation

To separate cytosolic and nuclear fractions, NE-PER nuclear and cytoplasmic extraction reagents (CER: Thermo Scientific #78833, Waltham, MA, USA) were employed. Upon completion of the treatment, A549 cells were collected in ice-cold 1 × PBS and centrifuged at 13,000× *g* for 15 min. A portion of the resulting cell pellet (20 μL) was mixed with ice-cold CER I (200 μL), CER II (11 μL), and protease inhibitors, and then vortexed and centrifuged to obtain the cytoplasmic protein extract (supernatant). The remaining pellet, containing the nuclei, was re-suspended in ice-cold NER, vortexed, and centrifuged to obtain the nuclear extract. Using the appropriate antibodies, the resulting fractions were subjected to immunoblotting analysis. The tubulin protein and lamin B protein were used as markers for the cytosol and nucleus, respectively.

### 2.8. Monolayer Wound Healing Assay

A549 cells were seeded in 6-well plates at a density of 1.5 × 10^5^ cells per well and incubated for 24 h. Subsequently, the cells underwent various treatments, including plasmid DNA transfection, siRNA transfection, or LPS (5 μg/mL) stimulation for 24 h. To inhibit cell proliferation during migration, serum starvation was implemented for 12 h before creating the “wound” scratch. Once the cell monolayer reached confluence, a 200 µL sterile plastic pipette tip was used to create scratches. The wounds were regularly photographed (0, 24, and 48 h) and the area of cell-free wounds was measured using a microscope (Axiovert 200, Zeiss, Baden-Wurttemberg, Germany). The assay was performed in at least three independent experiments.

### 2.9. Chromatin Immunoprecipitation (ChIP) and PCR

The ChIP experiment was conducted by following the protocol provided by Abcam (Cambridge, UK). In brief, A549 cells were subjected to LPS (5 μg/mL) exposure for 24 h, followed by the addition of formaldehyde as a crosslinker (final concentration: 0.75%) for 1 h. The crosslinking reaction was halted by treating with 125 mM glycine for 20 min. Subsequent to sonication in the ChIP lysis buffer (RIPA buffer), the fragmented chromatin protein complex was exposed to specific antibodies and subsequently precipitated using protein A/G beads. Afterward, the beads underwent washing, and the bound DNAs were eluted using elution buffer (containing 1% SDS and 100 mM NaHCO_3_). Subsequent removal of RNA and proteins was achieved by incubation with RNAse and proteinase K. Ultimately, DNA purification was carried out through phenol–chloroform extraction. PCR primers for ZEB1 (located on human chromosome 10, positions 31,318,016–31,319,215: forward primer: 5′-CAAACCTGCCCTTCCCCTCA-3′, reverse primer: 5′-CTCTACGGCCGGAACCTTGT-3′) and Snail1 (located on human chromosome 20, positions 48,598,513–48,599,512: forward primer: 5′-GAACGGGTGCTCTTGGCTAGCTG-3′, reverse primer: 5′-TCGAGCGAAGCGAGGCCTC-3′) were synthesized by Bioneer (Daejeon, Republic of Korea).

### 2.10. Immunofluorescent Staining

A549 cells were cultured and fixed with 4% paraformaldehyde for 10 min. To facilitate membrane permeabilization, a solution of 0.5% TX-100 detergents and PBS (1 × PBS) was applied for 10 min, followed by rinsing with 1 × PBS. The cells were then incubated overnight at 4 °C with primary antibodies targeting specific antibodies (1:100 dilutions). After incubation, the cells underwent washing with 1 × PBS. To visualize the antibodies, an Alexa Fluor 546 (red color)- or Alexa Fluor 488 (green color)-conjugated secondary antibodies were used at a dilution of 1:50 for 2 h at room temperature while keeping the environment free from light. The nuclei were stained with DAPI (4′,6-diamidino-2-phenylindole), which was added 10 min before washing with 1 × PBS at a concentration of 1 μg/mL. Fluorescence images were captured using a conventional fluorescence microscope (Axiovert 200, Zeiss, Oberkochen, Germany) and a confocal microscope (LSM 780NLO).

### 2.11. Quantitative Real-Time Polymerase Chain Reaction (qPCR)

Total RNA extraction from A549 cells was carried out using TRIzol reagent (Ambion, CA, USA). The concentration and purity of the isolated RNA were assessed spectrophotometrically at 260 nm and 280 nm with NanoDrop (Thermo Scientific). Reverse transcription of RNA into cDNA was performed using M-MLV reverse transcriptase (NEB-UK, Hitchin, UK) according to the manufacturer’s protocol. Subsequently, the generated cDNA was utilized with the Taq SYBR Rox 2X fast Q-PCR master mix (TQ1210, SMOBIO Technology, Inc., Hinchu, Taiwan) for real-time qPCR (RT-qPCR) quantification employing an Applied Biosystems StepOnePlus PCR system. Each sample was analyzed in triplicate, and the relative quantities of specifically amplified cDNA were determined using the comparative threshold cycle (C^T^) value method. GAPDH served as an endogenous reference gene. Primers for ZEB1 (forward: 5′-GTGCACAAGAAGAGCCACAAGTA-3′, reverse: 5′-GGTTGGCACTTGGTGGGATTAC-3′) and GAPDH (forward: 5′-AGAAGGCTGGGGCTCATTTG-3′, reverse: 5′-AGGGGCCATCCACAGTCTTC-3′) in measuring mRNA levels were synthesized by Bioneer (Daejeon, Republic of Korea).

### 2.12. Statistical Analysis

Data analysis and visualization were performed using GraphPad Prism 8 (GraphPad Software, San Diego, CA, USA). All experiments were repeated at least three times. The data are presented as mean ± SE (standard error). Statistical comparisons were performed using Student’s *t*-test with the GraphPad Prism software, and differences between two groups were considered statistically significant if the *p* values were below the designated threshold (* *p* < 0.05, ** *p* < 0.01, *** *p* < 0.001). 

## 3. Results

### 3.1. LPS Induces Superoxide Production through ROCK2 and p-p47phox

In this study, we have observed that LPS induces superoxide generation in the A549 lung cancer cell line in a concentration-dependent manner (Figure 1A). The LPS-mediated superoxide production was inhibited by Tat-C3, a Rho inhibitor, and Y27632, a ROCK inhibitor, highlighting the involvement of Rho and ROCK in regulating superoxide formation in response to LPS (Figure 1B). To distinguish ROCK types, we investigated the effects of ROCK1 or ROCK2 knockdown with siRNAs on superoxide production. Si-ROCK1 did not interfere with superoxide production, while si-ROCK2 prevented its generation in A549 cells, suggesting that ROCK2 primarily plays a role in superoxide production induced by LPS (Figure 1C). Consistent with these results, si-ROCK2, but not si-ROCK1, reduced the phosphorylation at the Ser345 residue of p47phox (Figure 1D). Of particular interest, reagents reducing ROS levels, namely apocynin, NAC, and BHC, subsequently reduced p-p47phox levels, suggesting that ROS induces the phosphorylation of p47phox and, consequently, NADPH oxidase activity (Figure 1E).

### 3.2. LPS Upregulates p-Tyr42 RhoA and Phospholipase D1 (PLD1) Levels with Forming a Protein Complex between p-Tyr42 RhoA and PLD1

LPS significantly increased both p-Tyr42 RhoA and PLD1 levels in concentration- and time-dependent manners (Figure 2A,B, respectively). Building on prior research indicating the interaction between RhoA-GTP and PLD1, leading to PLD1 enzyme activation, we investigated whether p-Tyr42 RhoA interacts with PLD1. In response to LPS, PLD1 co-immunoprecipitated with p-Tyr42 RhoA, indicating an interaction between PLD1 and p-Tyr42 RhoA (Figure 2C). Remarkably, the knockdown of RhoA with siRNA led to a reduction in PLD1 enzyme activity, while reconstitution with RhoA WT and the RhoA Y42E phospho-mimetic form restored its activity. Notably, the RhoA Y42F dephospho-mimetic did not restore PLD1 activity, emphasizing the critical role of the p-Tyr42 residue in RhoA for PLD1 enzyme activity (Figure 2D). Next, we investigated whether the p-Tyr42 residue of RhoA is pivotal for superoxide generation. RhoA knockdown with siRNA and reconstitution with the RhoA Y42F dephospho-mimic form suppressed superoxide production, whereas reconstitution of RhoA WT and the Y42E phospho-mimic form restored its generation (Figure 2E). Additionally, agents that mitigate ROS levels attenuated the levels of p-Tyr42 RhoA and PLD1 (Figure 2F). The noteworthy observation was that PLD1 co-immunoprecipitated with both RhoA WT and RhoA Y42E, a phospho-mimicking variant, while no interaction was detected with RhoA Y42F, a dephospho-mimicking variant. (Figure 2G). This implies that the phosphorylation of Ty42 in RhoA plays a pivotal role in this interaction with PLD1. 

### 3.3. p-Tyr42 RhoA Is Involved in the EMT Process and Cell Migration

LPS significantly promoted cell migration (Figure 3A) and induced changes in the expression of EMT marker proteins, specifically a decrease in E-cadherin and an increase in N-cadherin, ZEB1, and Snail1 (Figure 3B). Furthermore, RhoA knockdown with siRNA led to a marked reduction in the expression of N-cadherin and ZEB1, coupled with an increase in E-cadherin levels (Figure 3C). Notably, the reconstitution of the RhoA Y42F dephospho-mimetic form reversed the EMT process with an increase in E-cadherin expression and a decrease in ZEB1 and Snail1 levels, underscoring the critical role of the p-Tyr42 residue of RhoA in the EMT process triggered by LPS (Figure 3D). Additionally, it is noteworthy that transfection RhoA Y42F significantly decreased PLD1 levels (Figure 3D). Consistent with these findings, si-RhoA and the reconstituted RhoA Y42F, which mimic the dephosphorylated state, effectively prevented cell migration, whereas reconstitution with RhoA WT and the RhoA Y42E phospho-mimic form restored migratory behavior (Figure 3E).

Furthermore, si-RhoA also completely abolished PLD1 expression, and MG132 failed to restore PLD1 levels (Figure 3F). These findings suggest that p-Y42 RhoA may play a role in regulating PLD1 expression rather than protecting PLD1 from degradation. 

### 3.4. p-Tyr42 RhoA and PLD1 Bind to the Promoter of ZEB1, Influencing Cell Migration

Based on our earlier findings demonstrating that p-Tyr42 RhoA activates PLD1 activity (Figure 2D), we hypothesized that PA generated by activated PLD1 might play crucial physiological roles within cells. To explore this, we investigated the effect of PA on superoxide production. While LPS induced superoxide generation, the knockdown of PLD1 with siRNA significantly suppressed its production (Figure 4A). Furthermore, our results consistently showed that PA stimulated superoxide generation (Figure 4B). Interestingly, PA also increased the phosphorylation of Tyr42 on RhoA (Figure 4C). To corroborate these observations, we assessed the impact of PLD1 knockdown, and found that siPLD1 significantly attenuated LPS-induced cell migration (Figure 4D). Additionally, given the strong connection between cell migration and EMT, we investigated the expression of EMT marker proteins. LPS induced a decrease in E-cadherin and an increase in N-cadherin, ZEB1, and Snail1. Conversely, PLD1 depletion with siPLD1 led to an increase in E-cadherin and a decrease in N-cadherin, ZEB1, and Snail1, underscoring the pivotal role of PLD1 in the EMT process triggered by LPS (Figure 4E). Notably, PA (5 μM) mimicked the effects of LPS on the regulation of EMT marker proteins, leading to a decrease in E-cadherin and increases in N-cadherin, ZEB1, and Snail1 (Figure 4F). These results suggest that PA is involved in the regulation of the EMT process and, indeed, we demonstrated that PA promoted cell migration (Figure 4G). Building on previous reports suggesting that p-Tyr42 RhoA translocates to the nucleus [19,20], we examined the distribution of p-Tyr42 RhoA and PLD1 in response to LPS. Western blotting of nuclear and cytosolic fractions revealed that LPS induced an increase of both p-Tyr42 RhoA and PLD1 in the nuclear fraction compared to the cytosolic fraction (Figure 4H). Furthermore, immunohistochemical images confirmed that LPS elevated the levels of p-Tyr42 RhoA and PLD1 within the nucleus in A549 cells, with a portion of these proteins co-localizing (Figure 4I). Significantly, ChIP-PCR using antibodies against p-Tyr42 RhoA and PLD1 and ZEB1 promoter primers indicated that LPS increased the association of p-Tyr42 RhoA and PLD1 with the ZEB1 promoter, but not that of Snail1 (Figure 4J,K, respectively). Additionally, si-PLD1 markedly decreased ZEB1 mRNA levels in response to LPS (Figure 4L). Moreover, both si-RhoA and RhoA Y42F significantly attenuated ZEB1 mRNA levels in response to LPS (Figure 4M). These results confirm that p-Y42 RhoA and PLD1 contribute to the transcriptional regulation in ZEB1. 

### 3.5. MYH9 Is a PA-Binding Protein and Forms a Complex with p-Tyr42 RhoA and PLD1 

As a result, our objective was to investigate and identify the proteins that interact with phosphatidic acid (PA) in the downstream signaling pathway activated by LPS. We utilized PA-conjugated beads to capture proteins from the lysates of the A549 cells stimulated with LPS. This approach led to the identification of MYH9, also referred to as Myosin IIA, non-muscle myosin IIA (NMIIA), and ATP synthase β as the protein bound to the PA-conjugated beads (Figure 5A). Notably, MYH9 (Myosin IIA) was observed to increase in response to LPS (Figure 5B,C), and exhibited localization in both the cytosol and the nucleus, with a preference for the cytosol (Figure 5C). Subsequently, we noted an LPS-induced elevation of MYH9 in the nucleus (Figure 5C). Subsequently, we investigated the effect of MYH9 on cell migration, and found that the knockdown of MYH9 with siRNA suppressed cell migration (Figure 5D). Moreover, MYH9 co-localized with F-actin, and the levels of both the MYH9 and F-actin in the nucleus were elevated in response to LPS (Figure 5E). Previously, myosin IIA (MYH9) has been documented to be associated with cell migration, metastasis, and the EMT process [21]. In line with these results, the knockdown of MYH9 with siRNA led to an increase in E-cadherin expression and a decrease in ZEB1 and Snail1 levels (Figure 5F). To confirm this result, we determined the level of ZEB1 mRNA. Si-MYH9 significantly prevented ZEB1 mRNA levels (Figure 5G). Notably, we observed MYH9 binding to the ZEB1 promoter via ChIP-PCR, regardless of LPS stimulation (Figure 5H). This result suggests that MYH9 may be involved in the regulation of ZEB1 expression. Furthermore, in lysates of A549 cells stimulated by LPS, we observed the association of PA-conjugated beads with MYH9, PLD1, and p-Tyr42 RhoA (Figure 5I). Consistent with this result, PLD1 co-immunoprecipitated with p-Y42 RhoA and MYH9, p-Y42 RhoA co-immunoprecipitated with MYH9 and PLD1, and MYH9 co-immunoprecipitated with PLD1 and p-Y42 RhoA (Figure 5J). These results indicate that PA/MYH9/PLD1/p-Y42 RhoA form a complex in response to LPS. 

## 4. Discussion

This study was initiated with the question of whether p-Tyr42 RhoA plays a regulatory role in PLD1 enzyme activity, given the well-established role of RhoA-GTP in activating PLD1. Recently, researchers elucidated the crystal structure of PLD1 in association with RhoA-GTP. In this study, they discovered that the C-terminal domain of PLD1 binds to the switch I domain of RhoA-GTP [15]. Notably, the Tyr42 residue of RhoA is situated in switch I, although it is located in the marginal region of switch I (see Appendix A). Both RhoA and p-Tyr42 RhoA were found to interact with PLD1 (Figure 2C). However, it is worth mentioning that RhoA Y42F, which serves as a dephospho-mimic form, did not enhance PLD1 enzyme activity (Figure 2D). These findings substantiate our hypothesis that the p-Tyr42 residue of RhoA plays a critical role in the activation of effector proteins, such as PLD1, which leads to the generation of PA. 

Numerous proteins have been previously documented to bind to PA [22,23,24,25]. The PX domain of p47phox features two distinct basic pockets on its membrane-binding surface, each designed for specific phospholipids, including PI(3,4)P_2_/PI(3)P and PA/PS (phosphatidylserine). It is noteworthy that PA has been shown to enhance NADPH oxidase activity [26]. Similar findings were reported by another researcher, indicating that both PLD1 and PA stimulate ROS production, with PA binding and promoting NADPH oxidase activity [27]. Consequently, we can deduce that p-Tyr42 RhoA/PLD1/PA collectively play a critical role in superoxide generation. Furthermore, our previous research has unveiled a reciprocal relationship between superoxide and Tyr42 phosphorylation in RhoA, specifically involving the pathway: superoxide-Src–p-Tyr42 RhoA [20]. Additionally, we have found that p-Tyr42 RhoA activates ROCK2, which further phosphorylates and activates p47phox, resulting in the activation of NADPH oxidase and generation of superoxide: p-Tyr42 RhoA–ROCK2–p-p47phox–NADPH oxidase-superoxide [18]. Based on these findings, we propose that superoxide and p-Tyr42 RhoA mutually regulate each other in a positive feedback loop manner. Notably, we hypothesized the existence of another positive feedback loop pathway, involving the signaling cascade: PA-NADPH oxidase-superoxide-Src-p-Tyr42 RhoA-PLD1-PA (Figure 4C). 

PA serves as a pivotal signaling molecule, contributing to a diverse range of regulatory functions within cellar processes. These functions encompass cell metabolism and growth, cell death, cytoskeletal remodeling, exocytosis, receptor endocytosis, membrane trafficking, and organelle dynamics [23]. Remarkably, alterations in PA levels have been linked to changes in actin filament dynamics. The previous studies have reported that exogenous PA and PLD can induce stress fibers and increase F-actin levels [28,29]. An increase in the PA level is correlated with enhanced actin filament formation, while a decrease is associated with the disassembly of actin filaments [30]. We further verified that PA beads exhibit a robust binding affinity to actin (Figure 5A). Moreover, F-actin has been documented to stimulate PLD activity, whereas monomeric G-actin exerts an inhibitory effect on PLD activity [30]. 

To the best of our knowledge, this study represents the first observation of the binding between PA and MYH9 (Myosin IIA). As a result, we postulate that PA plays a critical role in stimulating the interaction between F-actin and MYH9 (Myosin IIA). MYH9 (Myosin IIA) is composed of a heavy chain (230 kDa) and two regulatory light chains (20 kDa) that control the myosin activity, along with two essential light chains (17 kDa) that stabilize the heavy-chain structure [31]. Phosphorylation of the C-terminal region of MYH9 by PKC, casein kinase II (CKII), and transient receptor potential melastatin 7 (TRPM7) plays a pivotal role in its assembly–disassembly process [31]. Given that we identified the binding of the MYH9 heavy chain, excluding the light chains, to PA-conjugated beads (Figure 5A), we postulate that the globular N-terminal region of MYH9 may contain the binding site for PA. Incidentally, it is well established that myosin light-chain kinase (MLCK) and ROCK phosphorylate regulatory subunit in MYH9 (Myosin IIA), thereby activating it [31]. Expanding on these discoveries, we suggest that MYH9 (Myosin IIA) binding to PA is activated simultaneously by p-Tyr42 RhoA/ROCK in both the cytosol and the nucleus. Actually, MYH9 and F-actin were co-localized in response to LPS, suggesting that MYH9 and F-actin interact with each other (Figure 5G). 

If PLD1 produces PA within the nucleus and binds to MYH9 (Myosin IIA), it prompts the question of the function of MYH9 (Myosin IIA) within the nucleus. Remarkably, the phosphorylation of Ser1916 in the α-helical rod and Ser1943 in the non-helical tail in MYH9 has been observed to increase during the TGF-β-induced EMT process [32]. Notably, PA can stimulate a variety of kinases, including Raf-1, PKCs, PKN, mTOR, mTORC2, Akt, PAK1, p70S6K1, Fer, GRK, LATS1, and KSR1 [33]. Based on these findings, we suggest that the C-terminal phosphorylation of MYH9 may be implicated in the EMT process. During the EMT process, key EMT transcription factors such as SNAI1 (Snail 1), SLUG (Snail 2), TWIST, and ZEB1/2 bind directly to the promoter region of E-cadherin, inhibiting its transcription. This results in a decrease in E-cadherin levels, which play a crucial role in the formation of adherence junctions, a critical factor for cell–cell interaction. As a consequence, the decrease in E-cadherin prompts cell migration and the initiation of metastasis. Notably, the downregulation of E-cadherin and concurrent upregulation of N-cadherin during EMT have been observed in various cancers. This cadherin ‘switch’ is associated with an enhancement in cell migration and invasion, contributing to the progression of cancer [34]. 

In this study, we investigated the localization of both p-Tyr42 RhoA and PLD1 in the nucleus (Figure 4I). Moreover, we found that p-Tyr42 RhoA and PLD1 possess the ability to bind to the promoter region of ZEB1 (Figure 4J). Furthermore, through co-immunoprecipitation (Figure 5J) and PA-conjugated beads, we observed an interaction between p-Tyr42 RhoA, PLD1 and MYH9 (Figure 5I). Additionally, it is worth noting that ROCK2 is present in the nucleus and has the capability to phosphorylate and activate p300 acetyltransferase enzyme activity. This suggests that ROCK2 may potentially play a role in the regulation of histone acetylation and influence specific gene expression [35]. Moreover, MYH9 (Myosin IIA) was found to be present in the nucleus (Figure 5B,C,E) and to regulate the expression of EMT marker proteins (Figure 5F). In addition to its conventional role as a motor protein, myosin exhibits increasing evidence of involvement in the regulation of the chromatin structure, chromosomal translocation, transcription regulation, and DNA repair [36]. Nuclear myosin I (NMI) and actin influence the chromatin dynamics and function in the interphase nucleus [37]. NMI interacts with transcription factors, while actin engages with RNA polymerase. Furthermore, NM1 contributes to the activation of RNA polymerase II transcription [38]. Additionally, NMI–actin interactions contribute to the transition of the initiation complex into the elongation complex [39]. Moreover, Myosin VI and Myosin V are implicated in the regulation of gene transcription. Notably, non-muscle Myosin II is present in the nucleus and facilitates the regulation of gene expression [40,41]. Building upon these prior findings, our hypothesis suggests that the p-Tyr42 RhoA/PLD1/MYH9 complex may possibly regulate ZEB1 expression in the nucleus (Figure 4J and Figure 5I,J). Moreover, we theorize that the function of MYH9 in the cytosol is to enhance its interaction with F-actin through p-Tyr42 RhoA/PLD1/PA, ultimately promoting cell migration. 

Nevertheless, several questions remain unanswered. The most significant among them is whether p-Tyr42 RhoA, PLD1, and PA generated by PLD1, in conjunction with MYH9 and actin, which interact with PA, play a role in regulating the cytoskeletal dynamics within the nucleus to influence the chromatin structure and gene expression. This particular aspect has not been investigated in the current paper and should be considered for future research endeavors. Furthermore, it is particular intriguing to identify the specific genes that are regulated by MYH9, p-Tyr42 RhoA, and PLD1 through transcriptome analyses using si-MYH9, si-PLD1, and RhoA Y42F transfection techniques. Overall, we have presented compelling evidence that p-Tyr42 RhoA plays a significant role in the transcriptional regulation of specific genes in conjunction with other transcription factors [7,19,20]. 

## 5. Conclusions

The inflammatory agent LPS induces the generation of superoxide, leading to phosphorylation at the Tyrosine42 of RhoA. Subsequently, p-Tyr42 RhoA activates PLD1, resulting in the production of PA. PA then binds to MYH9 (myosin IIA) and actin, regulating cell migration. Concurrently, p-Tyr42 RhoA, PLD1, and MYH9 bind to the promoter of ZEB1, a typical EMT marker protein, influencing ZEB1 expression and facilitating cell migration. This represents a novel mechanism for promoting cancer cell progress through inflammation. 

## Figures and Tables

**Figure 1 biomolecules-14-00006-f001:**
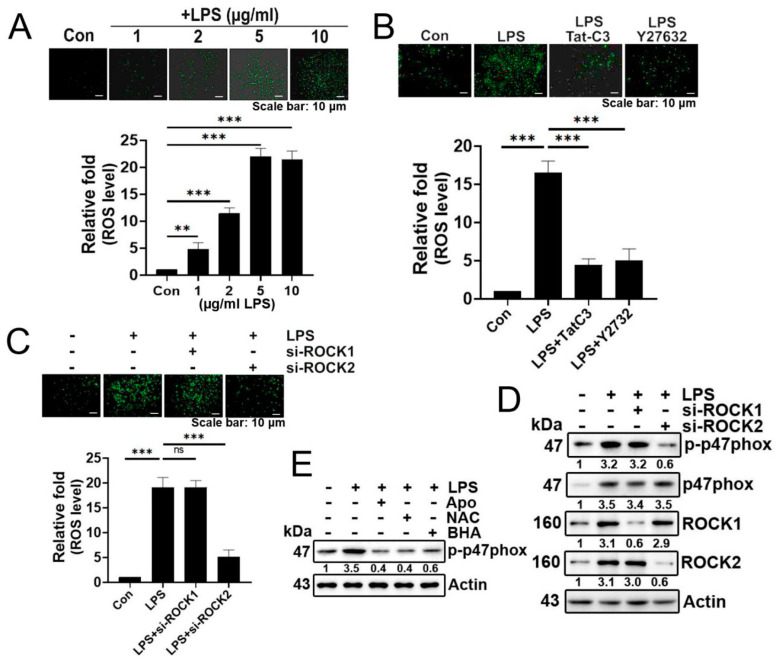
LPS increases superoxide generation via ROCK2 and p-p47phox. (**A**) Representative images depict the measurement of cellular ROS/superoxide production in A549 cells using the fluorescent probe 2′,7′-dichlorofluorescin diacetate (DCFDA) after LPS treatment at various concentrations (0, 1, 2, 5, 10 μg/mL) for 24 h. Produced superoxide was visualized with 10 μM (DCFDA) for 20 min, and detected using a fluorescence microscope (Axiovert 200, Carl Zeiss). (**B**) A549 cells were subjected to an 8 h period of serum starvation, followed by pretreatment with Tat-C3 (1 μg/mL) and Y27632 (10 μM) for 1 h. Subsequently, the cells were stimulated with LPS (5 μg/mL) for 24 h, and the measurement of reactive oxygen species (ROS) was performed using DCFDA. (**C**) Likewise, superoxide levels were assessed after the knockdown of ROCK1 and ROCK2 by si-RNA for 48 h, followed by the stimulation of LPS (5 μg/mL) for 24 h in A549 cells. (**D**) The protein levels of p-p47phox were determined through Western blotting in A549 cells. This was performed after 24 h of LPS (5 μg/mL) stimulation following the si-RNA-mediated knockdown of both ROCK1 and ROCK2. (**E**) A549 cells were pretreated with apocynin (1 μM), NAC (10 mM), and BHA (10 μM) for 1 h, followed by stimulation with LPS (5 μg/mL) for 24 h and the indicated proteins were detected by Western blotting. Adobe Photoshop (version 8) was used to quantify the band intensity of each of the representative photos of the Western blot. The data are mean ± SD of three independent experiments (**, *p* < 0.01, and ***, *p* < 0.001) in case without particular remark. Ns typically stands for non-significant, indicating that there is no significant difference or effect observed. Scale bars were indicated in the below each relevant figure. Western blot data are representative of at least three independent experiments.

**Figure 2 biomolecules-14-00006-f002:**
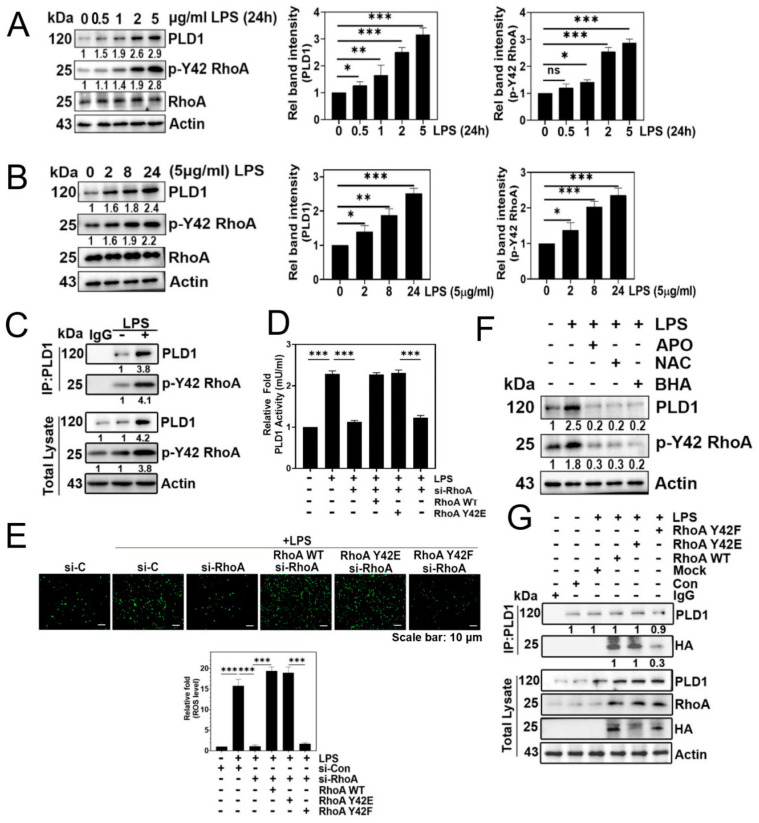
LPS upregulates p-Tyr42 RhoA, phospholipase D1 (PLD1) levels via generating superoxide. (**A**,**B**) Western blotting was employed to quantify the levels of RhoA, p-Y42 RhoA, and PLD1 in A549 cells treated with different concentrations (0–5 μg/mL) of LPS for 24 h (**A**), and a fixed concentration of 5 μg/mL for different time periods (0–24 h) (**B**). (**C**) Immunoprecipitation of PLD1 was performed in LPS-stimulated A549 cells to assess the interaction with p-Y42 RhoA using specific antibodies. (**D**,**E**) A549 cells, initially subjected to si-RhoA transfection (50 nM), underwent subsequent transfection with RhoA WT, Y42E, and Y42F plasmid DNA (2 μg) and were then stimulated with 5 μg/mL LPS for 24 h. The activity of PLD1 was determined using a colorimetric Phospholipase D activity assay kit at an optical density of 570 nm (**D**), while superoxide production was measured using DCFDA (**E**). (**F**) Prior to LPS stimulation (5 μg/mL) for 24 h, A549 cells were pretreated with apocynin (1 μM), NAC (10 mM), and BHA (10 μM) for 1 h. Western blotting was employed to examine changes in the levels of PLD1 and p-Y42 RhoA. (**G**) A549 cells were transfected with RhoA wild type as well as Y42E (phospho-mimic) and Y42F (dephospho-mimic form) for 24 h. Subsequently, cells were cultured in a serum-depleted media for 8 h and stimulated by LPS for 24 h. The interaction of PLD1 was then examined through immunoprecipitation using the HA probe. RhoA, PLD1, and HA protein levels were detected by Western blotting. The data are the mean ± SD of three independent experiments (*, *p* < 0.05; **, *p* < 0.01, and ***, *p* < 0.001) in case without particular remark. Scale bars were indicated in the below each relevant figure. Western blot data are representative of at least three independent experiments.

**Figure 3 biomolecules-14-00006-f003:**
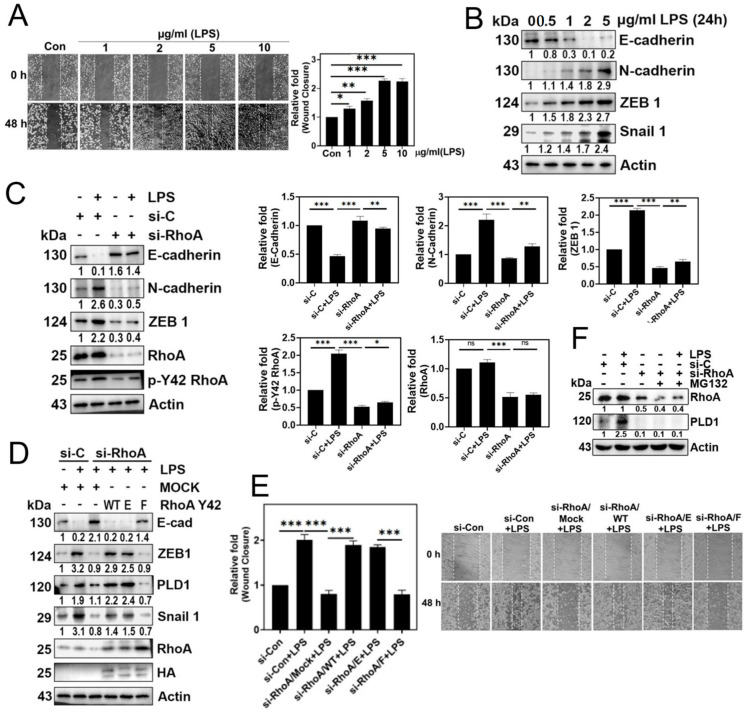
EMT and cell migration are facilitated by p-Tyr42 RhoA. (**A**) A549 cells were exposed to varying concentrations of LPS (0–10 μg/mL) for 48 h. Cell migration was evaluated at 0 and 48 h by observing under a microscope after scratching with a plastic pipette tip. (**B**) After treating A549 cells with LPS (0–5 μg/mL at different concentrations for 24 h), the expression levels of E-cadherin, N-cadherin, ZEB1, and Snail were examined through immunoblotting. (**C**) A549 cells were transfected with si-Con (20 nM) and si-RhoA (50 nM) for 48 h, followed by an 8 h starvation period before stimulation with LPS (5 μg/mL) for 24 h. Immunoblotting was employed to detect the expression of E-cadherin, N-cadherin, ZEB1, RhoA, and p-Y42 RhoA, which were subsequently plotted in a bar diagram. (**D**,**E**) After transfection with si-con (20 nM) and si-RhoA (50 nM) for 48 h, A549 cells underwent additional transfection with Mock, RhoA WT, Y42E, and Y42F plasmid DNA (2 μg) for 24 h. Subsequently, the cells were stimulated with 5 μg/mL LPS for 24 h. (**D**) Western blot analysis was employed to assess the levels of E-cadherin, N-cadherin, ZEB1, PLD1, Snail 1, HA, RhoA, and Actin as a loading control. (**E**) Cell migration was evaluated according to the methods described in (**A**). (**F**) A549 cells, transfected with si-con (20 nM) and si-RhoA (50 nM), were subjected to stimulation with LPS (5 μg/mL) in the presence of MG132 (1 μM) for 48 h. Changes in the levels of RhoA and PLD1 were assessed through Western blotting. The data are the mean ± SD of three independent experiments (*, *p* < 0.05; **, *p* < 0.01, and ***, *p* < 0.001) in case without particular remark. Ns typically stands for non-significant, indicating that there is no significant difference or effect observed. Scale bars were indicated in the below each relevant figure. The Western blot results are indicative of a minimum of three independent experiments.

**Figure 4 biomolecules-14-00006-f004:**
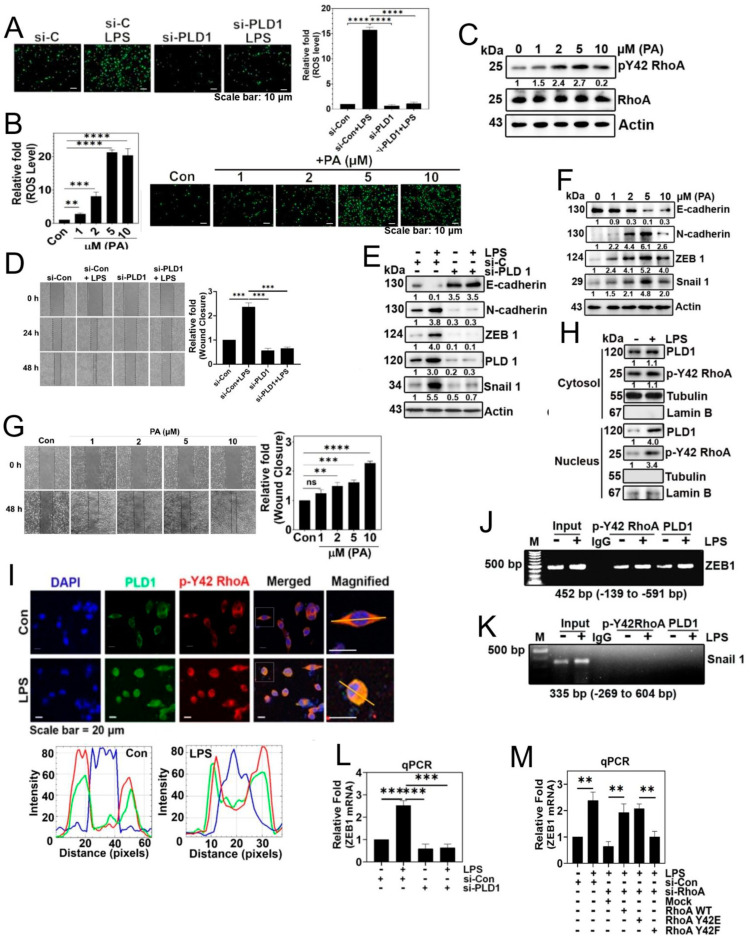
p-Tyr42 RhoA and PLD1 bind to the promoter of ZEB1, influencing cell migration. (**A**) A549 cells were transfected with si-Con (20 nM) and si-PLD1 (50 nM) for 48 h. Following 12 h of starvation, cells were stimulated with LPS (5 μg/mL) for 24 h to assess ROS generation using DCFDA, as depicted in Figure 1A. (**B**,**C**) A549 cells were activated with various concentrations of phosphatidic acid (PA) (0–10 μM) for 24 h. (**B**) ROS levels were then quantified as mentioned previously in Figure 1A. (**C**) Western blot analysis was performed to determine the expression levels of RhoA and p-Y42 RhoA. (**D**,**E**) A549 cells were transfected with si-Con (20 nM) and si-PLD1 (50 nM) for 48 h. After 12 h of starvation, cells were stimulated with LPS (5 μg/mL) for 24 h. (**D**) Cell migration rate was assessed according to Figure 3A. (**E**) Immunoblotting was conducted to determine the expression levels of E-cadherin, N-cadherin, ZEB1, PLD1, Snail 1, and Actin as a loading control. (**F**,**G**) A549 cells stimulated with various concentrations of PA (0–10 μM) for 24 h. (**F**) Immunoblotting was utilized to measure the expression levels of E-cadherin, N-cadherin, ZEB1, Snail 1, and Actin as a loading control. (**G**) Cell migration was measured as depicted in Figure 3A. (**H**) A549 cells were treated with LPS (5 μg/mL) for 24 h. Subsequently, cytosolic and nuclear fractions were isolated, and Western blotting was performed to identify PLD1 and p-Y42 RhoA. Lamin B and tubulin expression levels served as loading controls for the nuclear and cytosolic fractions, respectively. (**I**) A549 cells were cultured for 24 h, and following a 12 h period of serum starvation, the cells underwent stimulation with LPS (5 μg/mL) for 24 h. Subsequently, the cells were treated with 5 μg/mL of p-Y42 RhoA (Rabbit antibody) and 5 μg/mL of PLD1 (mouse antibody), and incubated overnight at 4 °C. The co-localization of PLD1 (green) and p-Y42 RhoA (red) was examined using confocal microscopy, with the nucleus labeled using 5 μg/mL DAPI (blue). White lines intersecting cells were used to measure the fluorescence intensity. (**J**,**K**) ChIP-PCR experiments were conducted on A549 cells exposed to LPS (5 μg/mL) for 24 h, using primers specific for ZEB1 (**J**) and Snail 1 (**K**), aiming to investigate the binding of p-Y42 RhoA and PLD1. (**L**) A549 cells were transfected with si-Con (20 nM) and si-PLD1 (50 nM) for 48 h. Following 12 h of starvation, cells were then stimulated with LPS (5 μg/mL) for 24 h. Subsequently, ZEB1 mRNA expression was quantified using qPCR. (**M**) Following transfection with si-con (20 nM) and si-RhoA (50 nM) for 48 h, A549 cells were subjected to additional transfection with Mock, RhoA WT, Y42E, and Y42F plasmid DNA (2 μg) for 24 h. Subsequently, the cells were stimulated with 5 μg/mL LPS for an additional 24 h. The expression of ZEB1 mRNA was evaluated using qPCR. The data are the mean ± SD of three independent experiments (**, *p* < 0.01; ***, *p* < 0.001; and ****, *p* < 0.0001) in case without particular remark. Scale bars were indicated in the below each relevant figure. Western blot data are representative of at least three independent experiments.

**Figure 5 biomolecules-14-00006-f005:**
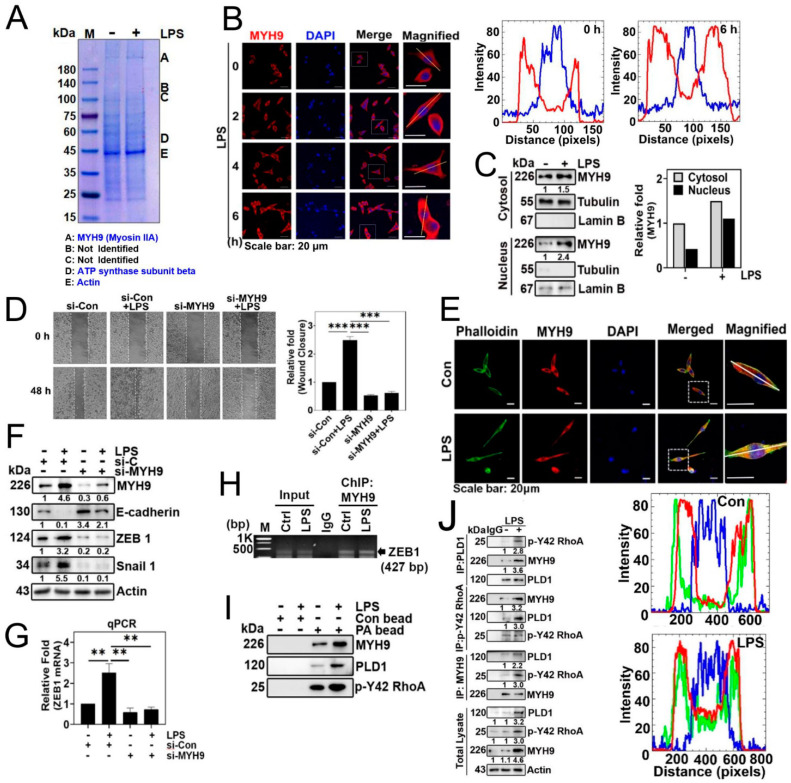
MYH9 is a PA-binding protein. (**A**) A549 cells were exposed to LPS (5 μg/mL) for 24 h to induce stimulation. Subsequently, the cell lysate was subjected to overnight incubation with phosphatidic acid (PA)-conjugated beads. The proteins were then separated using SDS-PAGE, and the identification of target protein species was performed through MALDI-TOF analysis. (**B**) A549 cells were plated for 16 h, and then treated with thymidine (2 mM) for 16 h for the first block of cell cycle arrest. Subsequently, the cells were rinsed with PBS and exposed to fresh media. The initiation of a second cycle arrest occurred by treating the cells with thymidine (2 mM) for an additional 16 h. Following the completion of the second arrest, the cells were then exposed to LPS (5 μg/mL), and the localization of MYH9 (in red) was observed at various time points (0–6 h) using confocal microscopy. White lines intersecting cells were used to measure the fluorescence intensity. (**C**) A549 cells were exposed to LPS (5 μg/mL) for 24 h. Following this, cytosolic and nuclear fractions were isolated, and Western blotting was conducted to detect MYH9 in both the cytosol and nucleus. Lamin B and tubulin expression levels were used as loading controls for the nuclear and cytosolic fractions, respectively. (**D**) A549 cells underwent transfection with si-Con (20 nM) and si-MYH9 (50 nM) over a 48 h period. Following an 8 h starvation of the cells, they were then exposed to LPS (5 μg/mL) for an additional 48 h. Cell migration was assessed at 0 and 48 h through microscopic observation. (**E**) A549 cells were grown in culture for 24 h. After a 12 h period of serum starvation, the cells underwent stimulation with LPS (5 μg/mL) for an additional 24 h. Subsequently, the cells were exposed to 5 μg/mL of MYH9 (rabbit antibody) and incubated overnight at 4 °C. Following this, MYH9 staining was conducted using Alexa Fluor-568 (red), and F-actin staining was performed using Alexa Fluor-488 Phalloidin, both at a dilution of 1:500 in PBS. The resulting localization of MYH9 (red), Phalloidin (green), and DAPI (blue) were examined through confocal microscopy. White lines intersecting cells were used to measure the fluorescence intensity. (**F**) A549 cells underwent transfection with si-control (20 nM) and si-MYH9 (50 nM) for a duration of 48 h. Following an 8 h period of starvation, the cells were exposed to LPS (5 μg/mL). Subsequently, alterations in the protein expression levels of MYH9, E-cadherin, ZEB1, and Snail 1 were assessed through immunoblotting. (**G**) A549 cells underwent transfection with si-Con (20 nM) and si-MYH9 (50 nM) for a period of 48 h. After a 12 h starvation phase, the cells were stimulated with LPS (5 μg/mL) for 24 h. Following this, the quantification of ZEB1 mRNA expression was carried out through qPCR. (**H**) A549 cells were grown and exposed to LPS (5 µg/mL) for 24 h. Immunoprecipitation was performed using the MYH9 antibody, with normal IgG serving as a negative control for the immunoprecipitation. A representative ChIP-PCR of ZEB1 was presented, utilizing a primer pair spanning the promoter region of ZEB1, obtained from the UCSC genome browser (genome.ussc.edu, Human GRCh37/hg19 chromosome 10: 31,318,016–31,319,215). (**I**) A549 cells were stimulated with LPS (5 μg/mL) for a duration of 24 h. Subsequently, the cell lysate underwent incubation overnight with both control beads and beads linked to phosphatidic acid (PA). Following this, protein separation was achieved through SDS-PAGE, and immunoblotting was employed to assess the protein expression levels of MYH9, PLD1, and p-Y42 RhoA. (**J**) A549 cells were cultured and treated with LPS (5 µg/mL) for 24 h. Immunoprecipitation of PLD1, p-Y42 RhoA, and MYH9 was conducted in both untreated and LPS (5 μg/mL)-stimulated A549 cells for 24 h to evaluate their interactions using specific antibodies. The data are the mean ± SD of three independent experiments (**, *p* < 0.01, and ***, *p* < 0.001) in case without particular remark. Scale bars were indicated in the below each relevant figure. Western blot data are representative of at least three independent experiments.

## Data Availability

The data presented in this study are available in insert article or Appendix A here.

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
