# Peer review of "Lipopolysaccharide Stimulates A549 Cell Migration through p-Tyr 42 RhoA and Phospholipase D1 Activity"

_biomolecules, 2023, doi:10.3390/biom14010006_

Round 1
Reviewer 1 Report (Previous Reviewer 2)
Comments and Suggestions for Authors
Thanks for throughly addressing the comments. Regarding comment #1, I believe the authors were referring to figure 5C, depicting the nuclear and cytosolic presence of MYH9. Main text body refers it correctly.
The revisions have notably enhanced the manuscript. In my opinion, the manuscript can now be considered for publication.
Author Response
Thank your constructive comments. We attached the file of response.

Reviewer 2 Report (Previous Reviewer 1)
Comments and Suggestions for Authors
Authors addressed most of our comments and strongly improved their manuscript. But two minor points need to be improved before the publication.
1, although authors discussed the positive feedback by introducing two parts in the discussion, they didn't link obviously this positive feedback with the current presentation data (Fig 2E conclusion). Could authors write the conclusions of previous studies to support a potential feedback loop that has been observed in figure 2 in this manuscript?
2, although authors discussed the nuclear myosin (different types of myosin) in the discussion, could authors also discuss the nuclear RhoA, nuclear ROCK, etc, in the discussion part? It will strongly improve this section.
Author Response
Thank your constructive comments. We attached the file of response.

Reviewer 3 Report (New Reviewer)
Comments and Suggestions for Authors
The manuscript titled “Lipopolysaccharide stimulates A549 cell migration through p- 2 Tyr 42 RhoA and Phospholipase D1 activity” suggests that p-Tyr42 RhoA and PLD1, involved in PA production, along with PA-bound MYH9, play a role in regulating ZEB1 expression, ultimately promoting cell migration. The research is interesting; however, I would suggest major revisions before accepting the manuscript for publication:
1. Some details in the experimental design sections are missing such as: concentrations of LPS used and the duration of exposure, concentrations of Tat-C3 and Y27632 inhibitors used for the inhibition experiments, concentration of PA used in the experiments and the duration of exposure.
2. In section 3.3, the authors should elaborate on the significance of the observed changes in EMT marker proteins (E-cadherin, N-cadherin, ZEB1, Snail1) induced by LPS and how these changes relate to the EMT process.
3. In section 3.4, the authors should discuss the mechanism underlying the observed increase in the phosphorylation of Tyr42 on RhoA in response to PA (Fig. 4C).
4. In section 3.5, the authors should talk about how the association of MYH9, PLD1, and p-Tyr42 RhoA with PA-conjugated beads was determined and offer insights into the potential functional implications of MYH9's localization in both the cytosol and nucleus, especially its elevation in the nucleus in response to LPS.
5. Discussion needs to be elaborated. Specifically,
· Provide additional details on the crystal structure of PLD1 in association with PIP2, PIP3, and RhoA-GTP. Explain how this structure relates to your findings regarding p-Tyr42 RhoA and PLD1 interaction.
· Elaborate on the functional significance of the Tyr42 residue of RhoA, especially in the context of its location in the marginal region of switch I and its role in activating PLD1.
· Discuss the broader implications of p-Tyr42 RhoA activation, particularly its connection to superoxide generation and the reciprocal relationship with superoxide. How does this activation influence other cellular processes?
6. Clearly outline the unanswered questions and propose future research directions. For example, explore whether the complex plays a role in regulating cytoskeletal dynamics within the nucleus and its potential influence on chromatin structure and gene expression.
7. At the end of the manuscript, figure captions are missing.
Author Response
Thank your constructive comments. We attached the file of response.

Round 2
Reviewer 3 Report (New Reviewer)
Comments and Suggestions for Authors
The manuscript reads fine now. I accept it for publication.
This manuscript is a resubmission of an earlier submission. The following is a list of the peer review reports and author responses from that submission.
Round 1
Reviewer 1 Report
Comments and Suggestions for Authors
Mahmud's manuscript provided an interesting observation that LPS can stimulate cell migration by Tyr42-Rho and PLD1 activity. But current studies missed some important key results that can strongly support some major conclusions. The biggest issue is how these 4 factors, PA, PLD1, RhoA and MYH9, all known without DNA binding domain, can bind the ZEB promoter to induce transcriptional of this key EMT regulator. Moreover, authors detected all expression levels and their changes by protein value, but not by mRNA; however, all these changes seem to go through transcriptional control of ZEB. Here, Q-PCR experiments are all missing for these key conclusions. Thus, all these defects prevent this current manuscript from being published at this journal.
Main comments:
1, in Fig2. E, RhoA can enhance ROS levels; differently, in Fig2. E, inhibition of ROS can also reduce the RhoA activity. It seems that RhoA and ROS might form a positive feedback loop. This feedback loop could be very important for this paper. Authors need to clarify this potential loop and its role for EMT and cell migration.
2, in Fig4I, it is weird to use HEK293T cells which seem not to support their major conclusion (no prominent difference between control and LPS treatment). Moreover, the crossline in Fig 4i and j as well as Fig5c-d should use the same standard; authors should cross the line along the long axis of cell, but not the short axis of cells (which will miss a lot of cytosolic region for a fair comparison).
3, it is very difficult to understand why authors compared MYH9 with p-LaminA/C. What is the correlation between transcriptional regulation and nuclear envelope regulation? Authors should clarify the reason why to use p-LaminA/C for this comparison.
4, It is unclear which subcellular regions is where PA, MYH9, PLD1, and p-Y42 RhoA can form a complex. Although authors performed the Co-IP in figure 5 for the data supporting this complex formation, we didn't agree that authors can use this complex in both cytosol and nucleus in the cartoon of fig5i. We cannot conclude that some components can be detected by Chip assay to see that they can form a complex. In addition, this complex directly play a role in controlling cell contraction for cell migration, or it can control ZEB for EMT to control cell migration. This is unclear and needs to be further clarified.
5, authors often confused the tumorigenesis (tumor formation) and invasion/metastasis (tumor changes from non-invasive to highly-invasive subtypes). Thus, a lot of conceptual mistakes occurred in the introduction and some other sections of text.
Some minor comments:
1, there are tons of mistakes in grammar and typo in almost all sections of manuscript.
2, some western blot results missed quantifications.
3, Fig 4 and 5 have some inconsistency between figure sequences and text sequences (such as Fig 4j, Fig 5d,f,g,h).
4, for readers without this background, it is unclear what is p47phpx and its ser345? Authors should explain this information in their text.
Comments on the Quality of English Language
A lot of mistakes in grammar and typo.
Reviewer 2 Report
Comments and Suggestions for Authors
The manuscript authored by Mahmud S et al. underscores the pivotal role of inflammation in instigating cancer cell migration while elucidating the underlying mechanism using a pulmonary epithelial cell line, A549, as a model. Although this article presents intriguing findings, it is marked by a lack of a coherent structure. To enhance the overall comprehensiveness of the study, the following points should be addressed:
1. It is advisable for the authors to also demonstrate the presence of MYH9 in both the cytosol and nucleus through Western blot analysis as well.
2. The manuscript mentioned that p-Y42 RhoA, PLD1 and MYH9 associated to form a complex. To substantiate this claim, it would be beneficial to include CO-IP blots from both cytosolic and nuclear fractions.
3. Inconsistencies between figure panels/legends and the text are observed at several junctures.
4. The manuscript mistakenly refers to PLD1 as PDL1 in various instances.
5. Figure 4, specifically panels A, D and E, includes the knockdown of PLD1, yet the legend incorrectly discusses the knockdown of RhoA.
6. Figure 4 would benefit from a clear explanation of the rationale behind the inclusion of HEK293 T cells alongside A549 to showcase the nuclear/cytosolic localization of PLD1 and p-Y42 RhoA.
7. In figure 5H, the quality of the actin loading control is subpar and should be improved.
By addressing these points, the manuscript can be enhanced in terms of its clarity, accuracy and overall impact.
Comments on the Quality of English LanguageModerate editing of the english language is required to rectify spelling errors and enhance coherency.